# Genetics of Female Pelvic Organ Prolapse: Up to Date

**DOI:** 10.3390/biom14091097

**Published:** 2024-09-01

**Authors:** Yuting Li, Zihan Li, Yinuo Li, Xiaofan Gao, Tian Wang, Yibao Huang, Mingfu Wu

**Affiliations:** 1National Clinical Research Center for Obstetrical and Gynecological Diseases, Department of Gynecology, Tongji Hospital, Tongji Medical College, Huazhong University of Science and Technology, Wuhan 430030, China; m202276373@hust.edu.cn (Y.L.); m202376637@hust.edu.cn (Z.L.); d202382265@hust.edu.cn (Y.L.); gaoxf@hust.edu.cn (X.G.); wangtian9@tjh.tjmu.edu.cn (T.W.); m202076343@hust.edu.cn (Y.H.); 2Key Laboratory of Cancer Invasion and Metastasis, Ministry of Education, Wuhan 430030, China

**Keywords:** pelvic organ prolapse, extracellular matrix, genetics, single-nucleotide polymorphisms, genome-wide association studies, epigenetic

## Abstract

Pelvic organ prolapse (POP) is a benign disease characterized by the descent of pelvic organs due to weakened pelvic floor muscles and fascial tissues. Primarily affecting elderly women, POP can lead to various urinary and gastrointestinal tract symptoms, significantly impacting their quality of life. The pathogenesis of POP predominantly involves nerve–muscle damage and disorders in the extracellular matrix metabolism within the pelvic floor. Recent studies have indicated that genetic factors may play a crucial role in this condition. Focusing on linkage analyses, single-nucleotide polymorphisms, genome-wide association studies, and whole exome sequencing studies, this review consolidates current research on the genetic predisposition to POP. Advances in epigenetics are also summarized and highlighted, aiming to provide theoretical recommendations for risk assessments, diagnoses, and the personalized treatment for patients with POP.

## 1. Introduction

Pelvic organ prolapse (POP) is a benign disease characterized by the descent of pelvic organs due to weakened pelvic floor muscles and fascial tissues, predominantly affecting elderly women. The organs involved include the anterior and posterior vaginal walls, the vaginal vault, and the uterus. This displacement often leads to the protrusion of neighboring organs into the vagina, resulting in cystocele, rectocele, and enterocele [1]. The primary symptoms are a sensation of a vaginal bulge or protrusion, vaginal heaviness, urinary and bowel symptoms, and sexual dysfunction, all of which can significantly impair a patient’s quality of life [2]. With an aging population, the medical burden associated with this disease is expected to rise. The prevalence of POP varies based on the population studied and the diagnostic criteria employed. A National Health and Nutrition Examination Survey conducted in America from 2005 to 2010 involving 7924 women over the age of 20 years reported a prevalence of symptomatic POP at 2.9%, with most patients being asymptomatic prior to diagnosis [3]. Conversely, a study from 2014 to 2016 involving 53,178 adult Chinese females identified a symptomatic POP prevalence of 9.6%, noting an increase with age [4]. Despite several identified risk factors, the pathogenesis of POP remains poorly understood. Recent research has increasingly focused on the genetic contributions to its pathogenesis. Therefore, this review summarizes recent advancements in understanding the pathogenesis and genetic predisposition to POP.

## 2. Materials and Methods

A comprehensive search of the published English language literature was conducted on PubMed. The search was performed using the MeSH terms combined with the following keywords: “pelvic organ prolapse”, “prolapse”, “genetics”, “polymorphism”, “SNP”, “genome wide association study”, “GWAS”, “exome sequencing”, “pathogenesis”, “molecular mechanism”, “muscle”, “nerve”, “extracellular matrix”, “cytokine”, “inflammation”, “senescence”, “aging”, and “epigenetic”. After the initial evaluation of titles and abstracts, we included original articles and reviews related to the genetic susceptibility and molecular mechanisms of POP pathogenesis. The references in the retrieved publications were hand-searched for relevant studies. Then, two authors independently extracted and made quality judgements on the data, and a third author was consulted in case of disagreement. The quality evaluation system mainly included relevance to genetic studies and molecular mechanisms of pathogenesis of POP, as well as methodological issues.

## 3. Pathogenesis of POP

Current research supports the notion that POP is a polygenic disorder influenced by a combination of genetic and environmental factors. Aging and vaginal delivery are deemed the most significant risk factors for the development of POP [5]. Consequently, POP may be regarded as a degenerative condition closely associated with tissue injury and impaired repair mechanisms [6]. Disruptions in pelvic floor support structures, such as tears in the levator ani muscles and the loss of apical support ligaments, along with alterations in the quantities and ratios of the extracellular matrix (ECM) can significantly affect the biomechanical properties of the tissues [7,8]. The maintenance of pelvic floor organs in their physiological positions is contingent not only upon the integrity of these support structures, but also on their functional normalcy. Pathophysiological states, such as oxidative stress (OS), mitochondrial dysfunction, protein homeostasis imbalance, and persistent inflammation, are apparent in the pelvic floor tissues and may contribute to compromised pelvic floor support [9,10,11]. This review discusses the potential pathogenesis of POP by focusing on both the pelvic floor muscles and ligamentous fascia, with an emphasis on the pathophysiological alterations occurring in both aging and injury states (Figure 1).

### 3.1. Damage in Pelvic Floor Muscles and Nerves

The predominant muscle of the pelvic floor, the levator ani, is a striated muscle that includes the pubococcygeus, iliococcygeus, and coccygeus muscles [12]. The continuous contraction of these slow-twitch fibers helps maintain the pelvic organs in their normal anatomical positions. In patients with POP, however, the levator ani muscle fibers are often fragmented, display a reduced diameter, decreased density, and are sparsely arranged [13]. Compared to controls, women with POP frequently exhibited defects in the levator ani muscles and produce less vaginal closure force during maximal contraction [14].

During vaginal delivery, pelvic floor tissues may suffer from ischemia and hypoxia, leading to immediate muscle damage. While this damage can gradually repair post-delivery, a significant portion remains improperly healed. Furthermore, chronic increased abdominal pressure may also contribute to persistent stress, exacerbating pelvic floor muscle damage. In vitro experiments have demonstrated that mechanical stress on myofibroblasts and levator ani muscles results in notable OS [15], triggering cellular mitochondrial dysfunction and impeding tissue repair. This eventually contributes to pelvic floor weakness [16]. These findings suggest that the inadequate repair of post-delivery levator ani muscle damage, coupled with chronic injury, are likely initial factors for POP.

The physiological function of muscle is contingent upon nerve innervation. In the absence of innervation, the differentiation and formation of mature muscle fibers are hindered, potentially leading to muscle atrophy due to the lack of neurotrophic support. Consequently, both the physiological function and structure of the muscle are compromised. Notably, denervation changes, including significant reductions or the absence of neuropeptide Y, vasoactive intestinal polypeptide, and substance P immunoreactive nerves, have been observed in the levator ani muscles of patients with POP [17]. However, previous studies have not identified any neuropathy in the levator ani muscles of POP patients [12], leaving the nerve injury hypothesis for the levator ani muscles open to question.

The degradation of muscle function inevitably occurs with aging. The disruption of protein homeostasis and mitochondrial dysfunction are key manifestations of muscle aging. This includes the decline of the heat shock protein family, autophagy-lysosome pathways, ubiquitin–proteasome systems, and mitochondrial quality control [18,19]. The role of mitochondrial disease and cumulative OS in the pathophysiology of pelvic floor disorders has been documented [20]. However, no studies have specifically addressed the aging of pelvic floor skeletal muscles. Further exploration into the mechanisms of aging in these muscles could enhance our understanding of the onset of POP.

### 3.2. Disorders of ECM Metabolism in the Pelvic Floor

The ECM comprises structural proteins such as collagen and elastin, along with matrix adhesion molecules including fibronectin, laminin, and proteoglycans. Primarily, collagen and elastic fibers are synthesized by fibroblasts. Factors inducing fibroblast apoptosis and dysfunction may weaken pelvic floor connective tissue by impacting ECM metabolism. On the other hand, an imbalance between proteolytic enzymes matrix metalloproteinase (MMPs) and the tissue inhibitor of matrix metalloproteinases (TIMPs) can also lead to excessive ECM degradation, contributing to the weakening of pelvic floor tissues observed in POP [21]. Recent studies highlight the influence of OS, mitochondrial dysfunction, hormone metabolism, and inflammation on fibroblast behavior.

#### 3.2.1. Oxidative Stress (OS)

The cytoskeleton in fibroblasts can sense mechanical stress and convert it into biochemical signals that are transmitted to the nucleus to regulate gene expression and protein translation [22]. In vitro studies have demonstrated that mechanical stress can lead to an accumulation of reactive oxygen species (ROS) in fibroblasts. Elevated levels of intracellular OS can induce cellular senescence and apoptosis and reduce the expression of type I collagen. During this process, the phosphorylation activation of the PI3K-Akt signaling pathway in prolapsed tissues increases the phosphorylation level of FOXO1 and inhibits its transcriptional regulatory activity, which in turn downregulates the expression of antioxidant proteins, such as glutathione peroxidase 1 (GPX1) and Mn-superoxide dismutase (Mn-SOD). This results in an imbalance between the reduced antioxidative capacity and the accumulation of intracellular ROS [11]. Markers of oxidative damage, such as 8-hydroxyguanosine (8-OHdG) and 4-hydroxynonenal (4-HNE), were significantly higher in POP patients compared to the controls. A further in vitro challenge of human USL fibroblasts using hydrogen peroxide (H_2_O_2_) revealed that H_2_O_2_ can regulate the cellular collagen metabolism in a dose-dependent manner, with small doses promoting collagen synthesis and higher doses enhancing collagen breakdown. Concurrently, levels of MMP2 and TIMP2 varied with the degree of OS [10]. Notably, in patients with POP, the overexpression of mitochondrial fusion 2 (MFN2) and advanced glycation end products mediate a reduction in the fibroblasts’ proliferative capacity, causing cell cycle arrest in the G0/G1 phase and the diminished production of type I and III procollagen via the Ras-Raf-ERK axis, RAGE and/or p-p38 MAPK, and NF-κB-p-p65 pathways, as documented in references [23,24,25]. This further suggests that OS may influence the ECM metabolism. Chronic long-term stress induces significant OS in pelvic floor tissues, potentially affecting their ECM metabolism.

#### 3.2.2. Mitochondrial Dysfunction

Pelvic floor tissues become hypoxic during labor or chronic increased abdominal pressure. Hypoxia induces the elevated expression of hypoxia-inducible factor 1α (HIF-1α) in the fibroblasts of the uterosacral ligament, resulting in mitochondrial dysfunction and the subsequent apoptosis of fibroblasts [26]. Mitochondrial DNA in the uterosacral ligament was increased in premenopausal patients with POP and significantly decreased in postmenopausal POP patients compared to the non-POP group. The elevated copy number of mitochondrial DNA may provide energy for tissue repair; however, this compensatory process was absent in postmenopausal patients. Furthermore, the expression of the transcriptional coactivator PGC-1α, a principal regulator of mitochondrial biogenesis, was upregulated in POP patients, potentially compensating for mitochondrial dysfunction by enhancing mitochondrial biogenesis in prolapsed tissues [27]. This evidence suggests that mitochondrial dysfunction plays a role in the development of POP.

#### 3.2.3. Hormonal Changes

The role of estrogen in the development of POP remains controversial. Predominantly, patients with POP are postmenopausal, suggesting that reduced estrogen levels may facilitate POP development. Estrogen regulates the expression of genes associated with the ECM metabolism [28,29], including lysyl oxidase-like-1 (LOXL1), an enzyme that catalyzes the deamination of lysine residues in protoelastin. These residues form cross-links to create stable elastin and promote covalent linkages between elastin and collagen [30]. Recent studies indicate that a stiff ECM can enhance the transformation of fibroblasts into myofibroblasts, leading to excessive collagen deposition, tissue contracture, and dysfunction. Conversely, estrogen impedes fibroblast differentiation by upregulating DNA methyltransferase1 (DNMT1) expression, thus inhibiting POP progression [31]. However, it has been observed that estrogen markedly suppresses fibroblast proliferation, and estrogen supplementation has not proven to be an effective treatment [32]. Clinical trial results also revealed that perioperative vaginal estrogen did not increase surgical success rates, suggesting that estrogen supplementation primarily mitigates symptoms related to vaginal atrophy with a limited therapeutic impact on POP [33]. In addition to an estrogen deficiency, other hormonal shifts including a variation in progesterone levels can influence ECM metabolism. Although findings on progesterone receptor (PR) expression levels are mixed, an increasing number of studies indicate elevated PR in patients with POP [34,35,36]. The research demonstrates that both estrogen and progesterone contribute to the integrity of pelvic floor support tissues by inhibiting MMP13 and collagenase production, thus reducing ECM degradation [37,38]. It is hypothesized that postmenopausal women experience a loss of this protective effect due to a hormone deficiency. Nonetheless, the exact role of progesterone remains unclear.

#### 3.2.4. Inflammation

Inflammatory cytokines such as IL-1, IL-6, and TNF-α have been implicated in the pathogenesis of POP through the promotion of ECM component breakdown. It has been observed that in patients with POP, the levels of pro-inflammatory cytokines IL-6 and TNF-α positively correlate with MMP1 and MMP2, while inversely correlate with TIMP1 [21,39]. Furthermore, IL-1β plays a role in the regulation of elastin expression [40]. Repeated mechanical stress, resulting from factors such as childbirth, heavy lifting, or chronic coughing, can upregulate genes associated with inflammation, subsequently leading to ECM degradation and pelvic floor dysfunction [38,41].

In conclusion, damage to the supporting tissues of the pelvic floor, including muscles, nerves, and ECM, is intricately involved in the pathogenesis of POP. Factors such as aging, continuous stress, and fluctuations in estrogen and progesterone contribute to the inevitable onset and progression of POP by promoting OS overproduction, mitochondrial dysfunction, the chronic inflammatory environment, and protein destabilization. Notably, current research has largely overlooked the role of pelvic floor muscles, whereas the impact of the ECM in pelvic floor tissues is increasingly recognized as more significant.

## 4. Genetic Studies of POP

Different populations and individuals exhibit varying predispositions to polygenic diseases due to distinct genetic structures and the influence of external environments. Notably, the prevalence of POP varies significantly among women of different races, suggesting that diverse genetic backgrounds may influence the etiology of POP [42]. Moreover, genetic variations in the number, structure, or function of pelvic floor connective tissue components across races may affect the incidence of POP.

Epidemiologic studies highlight the genetic predisposition to POP. The research indicates that sisters of patients with severe POP are five times more likely to develop the condition compared to women without a familial history [43]. A family history of POP is associated with a 2.7-fold increased risk (odds ratio (OR): 2.64; 95% confidence interval (CI): 2.07, 3.35) and a 1.4-fold increased risk of recurrence (OR: 1.44; 95% CI: 1.00, 2.08) [44]. Importantly, the aggregation of POP within families is not solely attributable to genetic factors; environmental influences, such as similar lifestyle habits and their socioeconomic status, also play a significant role. A comparative study of 3376 identical twins and 5067 fraternal twins found that identical twins had a higher concordance for developing POP. Quantitative genetic analyses suggest that genetic factors account for 43% of POP cases, while environmental factors contribute to 57%. Thus, genetic predisposition significantly influences the development of POP [45].

In recent years, the genetic susceptibility of POP has been partially elucidated through genomic studies, including candidate gene association studies, genome-wide association studies (GWAS), linkage analyses, and whole-exome sequencing (WES). The majority of these studies have concentrated on genes involved in the ECM metabolism in pelvic floor connective tissue. Numerous case–control studies have demonstrated that the abnormal composition and metabolism of connective tissue are linked to POP. Additionally, genetic knockout mouse models related to the elastic fiber metabolism, such as *LOXL1* [46,47] and fibronectin-5 (*FIBN5*), have exhibited severe POP symptoms, vaginal wall weakness, paraurethral lesions, and lower urinary tract dysfunction following pregnancy and vaginal delivery [5,48,49].

### 4.1. Linkage Analyses

Linkage analysis is a technique used to map the location of genes associated with specific traits or diseases by studying inheritance patterns of genetic markers. If family members affected by the same disease share a specific chromosome region, it is likely that the susceptibility gene for this disease is located in or near that region. The principles of genetic linkage are employed to genotype a reference locus within the family line and then mathematically determine whether the locus co-segregates with the disease, thereby exploring the relationship between causative genes and the reference locus [50]. Nikolova et al. first conducted a linkage analysis on POP family, where three generations of females exhibited symptoms of prolapse at a very young age. They identified ten linkage regions with potentially strong candidate genes involved in ECM metabolism, such as *LAMC1*, *LAMC2*, and *COL8A1*. Furthermore, they discovered that POP was inherited in an autosomal dominant manner with significant epistasis in this family [51]. Subsequently, Allen-Brady et al. performed a linkage analysis on 70 members from 32 families, each with at least two members suffering from moderate-to-severe POP. Their study identified a significant linkage of pelvic floor dysfunction susceptibility genes in the p21 region of chromosome 9 [52]. In a further linkage analysis involving 299 POP patients from 83 families, chromosomes 10q and 17q were found to be closely associated with the development of POP, indicating potential regions for susceptibility genes [53]. Future studies are needed to further narrow down these regions to pinpoint the gene loci associated with POP.

### 4.2. Candidate Gene Association Studies

Single-nucleotide polymorphisms (SNPs) are sites in DNA where individual nucleotides vary from person to person, representing the most abundant type of DNA sequence variation in the human genome. Only a small fraction of these SNPs are biologically significant and contribute to human genome diversity. SNPs can influence the promoter activity of DNA and pre-mRNA conformation, leading to alterations in amino acid sequences and protein function, thereby playing a direct or indirect role in phenotypic expression. Numerous studies have investigated POP-related SNPs in patients by DNA sequencing, targeting genes related to ECM synthesis and catabolism, including *COL1A1*, *COL3A1*, *LAMC1*, *LOXL1*, *FBLN5*, and *MMP*. Additionally, polymorphisms of ER may also be associated with POP (Table 1). However, these studies often face limitations such as small sample sizes and varying criteria for identifying cases and controls. Overall, no SNPs have yet been identified that strongly correlate with the development of POP.

#### 4.2.1. *COL1A1*

The most abundant ECM of pelvic floor connective tissue is composed of collagen types I and III [80]. Type I collagen is a heterotrimeric protein consisting of two α1 chains and one α2 chain, with the α1 protein chain encoded by *COL1A1* located on chromosome 17, q21.31–q22. The rs1800012 of *COL1A1* was found to be associated with an increased risk of stress urinary incontinence (SUI) [81]. The substitution of guanine with thymine (G/T) at the Sp1 binding site in *COL1A1′s* transcription factor leads to the abnormal production of the α1 chain, increasing the formation of homotrimeric proteins and decreasing connective tissue strength. Both POP and SUI are believed to share an etiological basis, prompting studies on the association between this polymorphism and POP. Feiner et al. reported an OR of 1.75 for the GT genotype prevalence of rs1800012 between POP cases and controls [57]. Subsequent studies in diverse populations such as Brazil, Korea, Italy, China, and Japan found no significant correlations [53,54,55,57,58,59]. However, Cartwright et al.’s meta-analysis identified an association between *COL1A1* rs1800012 (G→T) and POP (T vs. G, OR: 1.33, 95% CI: 1.02–1.73) [82]. Similarly, Allen-Brady et al. conducted a meta-analysis from 2015 to 2020, which also suggested an association between rs1800012 and POP [83]. Although individual studies may not have found significant associations, the data integration from global studies indicates a potential link between the mutant T allele and POP, though further validation in larger cohorts is necessary.

#### 4.2.2. *COL3A1*

*COL3A1*, located at chromosome 2, q24.3–q31, encodes the α1 protein chain of type III collagen. This fibrillar protein comprises three identical α1 chains, forming fibers that are finer, more sparsely arranged, and more elastic than type I collagen fibers, significantly contributing to tissue repair. In patients with POP, an increased presence of type III collagen at non-prolapse sites possibly indicates tissue repair after pelvic floor injury [84]. The polymorphism of *COL3A1* may alter the α1 protein chain, impacting the mechanical properties of type III collagen and affecting the structures supporting the pelvic floor. Chen et al. assessed the genetic association using a polymerase chain reaction and restriction fragment length polymorphism (PCR-RFLP) analysis in a study of 84 POP patients and 147 controls from Taiwan, discovering a significant association between the AA genotype of *COL3A1* (rs1800255) and POP. The genetic mutation in exon 30, from a guanine to an adenine (G→A), transforms the encoded alanine to threonine, potentially disrupting the triple-helical structure of type III collagen due to increased hydrophilicity [64]. Similarly, Kluivers et al. found an association between rs1800255 and POP in a study involving 202 patients and 102 controls [63]. Moreover, a meta-analysis up to December 2012 indicated a significant association of rs1800255 with POP in Asian and Dutch females, with an OR of 4.79 (95% CI: 1.91–11.98) [85].

All the aforementioned studies have utilized PCR-RFLP to draw their final conclusions. Additionally, Lince et al. confirmed the polymorphism of gene rs1800255 by employing the high-resolution melting curve (HRMC) technique. They observed differential expression between the case and control groups in Dutch women; however, it was not possible to demonstrate that this polymorphism was associated with POP. Consequently, they recommended that the conclusions of other association studies using PCR-RFLP should be approached with caution [62]. Subsequent studies involving Brazilian, Japanese, and Chinese females have all indicated that *COL3A1* rs1800255 is not linked to POP [54,55,56,61]. Nevertheless, a meta-analysis of data up to March 2021 suggests that rs1800255 may be a risk factor for POP in Caucasian populations. Moreover, Caucasian individuals carrying the A allele or AA genotype are at an increased risk of developing POP [86].

Jeon et al. conducted PCR-RFLP analyses on 36 POP cases and 36 normal controls, discovering that the G/A mutation at position +2092 of *COL3A1* exon 31 was associated with POP in Korean women, with an OR of 3.2 for POP in women carrying the G allele [66]. In contrast, a study in a Brazilian population found no association between this polymorphism in exon 31 of *COL3A1* and POP [65]. Thus, the research on *COL3A1* polymorphisms and POP has yielded inconsistent results. The connection between *COL3A1* polymorphism and POP remains inconclusive, influenced by factors such as the testing technology, racial differences, and the sample size.

#### 4.2.3. *LAMC1*

Laminin, a crucial non-collagenous component of the ECM glycoproteins, constitutes the basement membrane and plays a vital role in mediating cell adhesion, as well as regulating cell growth and differentiation. It is a heterotrimer composed of α, β, and γ chains and is present in at least 15 distinct forms. The gene *LAMC1*, which encodes the γ-chain, is located on chromosome 1 at q25.3 [67]. The altered laminin expression in tissues from patients with SUI suggests that changes in laminin may compromise the integrity of the basement membrane [87]. Previously, Nikolova et al. identified a candidate gene in vaginal tissues of POP-afflicted family through linkage and co-segregation analyses, discovering a mutation (C→T) in the *LAMC1* promoter region, rs10911193, which may heighten the risk of an early-onset POP [51]. Further research involving Caucasian and African–American females with a stage II-IV POP revealed racial differences in *LAMC1* polymorphisms, though no significant correlation was found between the POP stage and *LAMC1* variants [68]. Another study across diverse ethnic groups also found no association between *LAMC1* polymorphisms and POP [67]. Thus, the relationship between this gene polymorphism and POP requires additional validation across different racial groups.

#### 4.2.4. *LOXL*

Lysyl oxidase-like gene 1 (*LOXL1*), situated on chromosome 15 at q24.1, encodes a copper-dependent monoamine oxidase. Within the ECM, lysyl oxidase catalyzes the cross-linking of lysine residues in elastin and hydroxylysine residues in collagen, thereby stabilizing and forming an elastic meshwork [30]. Liu et al. knocked out *Loxl1* in a mouse model and observed that, in contrast to wild-type mice, *Loxl1*^−/−^ mice exhibited abnormal elastic fiber deposition in the uterus post-delivery and developed symptoms related to POP. This indicates that *LOXL1* is crucial in maintaining the structure and function of elastic fibers, and its genetic deficiency may foster the progression of POP [47]. Moreover, molecular analyses of anterior vaginal wall tissues from POP patients revealed the significantly reduced expression of LOXL1 and LOXL3, suggesting that LOXL defects could lead to abnormal elastic fiber remodeling in pelvic connective tissues, thereby contributing to the onset and exacerbation of POP. Nonetheless, case–control studies conducted with Japanese and Brazilian women have not demonstrated an association, and no SNPs in this gene have been linked to POP [54,69].

Furthermore, a linkage analysis revealed that chromosome 10 q24–26 is associated with POP. *LOXL4*, located on chromosome 10 q24.2, is identified as the most likely candidate gene in this region [53]. It plays a crucial role in connective tissue biogenesis by catalyzing the cross-linking between collagen and elastin. A study involving Brazilian women found no correlation between the *LOXL4* gene (rs2862296) and POP [70]. In contrast, this polymorphism was linked to the development of POP in Japanese women [54]. Given the limited research on the association between *LOXL4* polymorphisms and POP and the varying conclusions of different studies, further validation through more extensive research is necessary.

#### 4.2.5. *FBLN5*

Fibulin is a class of secreted glycoproteins distributed in the ECM. Fibulin5 (FBLN5) acts as an adaptor protein linking cross-linking enzymes with structural components of elastic fibers, such as elastin, fibrillin-1, and LOXL1. It organizes the assembly of elastic fibers on microfibrillar scaffolds and stabilizes the structure of the basement membrane [48]. Studies on *Fbln5^−/−^* mice have shown progressive symptoms of genital prolapse, indicating that FBLN5 may play a crucial role in maintaining the stability of the pelvic floor support structure. Furthermore, FBLN5 can suppress the expression of MMP9, which regulates the homeostasis of the pelvic floor’s ECM and prevents the occurrence of POP [48]. Khadzhieva et al. conducted the first study on *FBLN5* polymorphisms in patients with stage III-IV POP and 292 normal controls, finding significant associations with POP, especially in patients with a history of pelvic floor injury, at rs2018736 and rs12589592 [71]. Additionally, a meta-analysis from 2015 to 2020 indicated that the polymorphism rs12589592 in *FBLN5* was connected to POP risk with an OR of 1.46 and a 95% CI of 1.11–1.82 [83]. However, Paula et al. found no association between *FBLN5* polymorphisms and POP in Brazilian women [72].

#### 4.2.6. *MMPs* and *TIMPs*

MMPs, a class of proteases capable of degrading the ECM and basement membrane components, have their increased expression levels directly associated with the onset and progression of POP [88,89]. MMP1, also known as mesenchymal collagenase, targets type I, II, III, and IV collagens and is primarily located in genital tissues, the colon, and blood vessels. In Italian women, an insertion or deletion of a guanine at −1607/1608 upstream of the *MMP1* transcriptional start site has been linked with POP development [58]. Two polymorphic loci in POP patients were examined, including the *MMP1* promoter region −1607/−1608 G insertion or deletion and the *MMP3* promoter region −1612/1617 A insertion or deletion. Although no individual SNP was associated with POP, a combined analysis of these two SNPs indicated the higher expression of the gene polymorphism in patients with POP compared to controls [73].

MMP9, also known as type IV collagenase or gelatinase B, primarily expressed in blood vessels and pelvic tissues, is responsible for degrading collagen, proteoglycans, laminin, and fibronectin. Chen et al. reported significant associations of the AG and GG genotypes of MMP9 rs17576 with POP [76]. Additionally, two other polymorphic sites in *MMP9*, rs3918253 and rs3918256, have been linked to POP in non-Hispanic white women [75]. Furthermore, it has been observed that polymorphisms in *MMP10*, specifically rs17435959, may contribute to POP onset. This polymorphism involves a substitution of leucine with valine at amino acid 4 in the first exon of the gene. A case–control study indicated a significant difference in the frequencies of the rs17435959 G/C between the cases and controls [77].

TIMP serves as an inhibitor of endogenous MMPs and is crucial for maintaining the metabolic balance between ECM synthesis and degradation. Reduced TIMP expression levels in pelvic floor tissues have been directly linked with the onset and progression of POP [88]. Furthermore, a linkage analysis has identified chromosome 17q25 as potentially related to POP, with *TIMP2* being the most likely candidate gene in this region [53]. However, no studies currently explore the association between this gene polymorphism and the pathogenesis of POP.

In summary, research examining the relationship between polymorphisms in genes associated with ECM degradation and POP has produced inconsistent results. Further studies with larger sample sizes across diverse ethnic groups are necessary for additional clarification.

#### 4.2.7. *ER*

Estrogen receptors (ER) are categorized into two primary subtypes: ERα and ERβ. ERα is predominantly expressed in the uterus of adult women, whereas ERβ is mainly expressed in other estrogen-targeted tissues [90]. Both ERα and ERβ can influence the strength of pelvic floor supportive tissues by regulating the synthesis and degradation of the ECM. In patients with POP, the expression levels of ER in the uterine and USL are significantly reduced prior to menopause [91]. Chen et al. observed that the frequency of the *ERβ* haplotype CGCGC was elevated in women with POP (16.7%) [92]. Research conducted in Chinese and Israeli populations has demonstrated a significant association between the rs2228480 polymorphism in *ERα* and the development of POP, identifying both heterozygous and homozygous subtypes as risk factors [79]. Moreover, a meta-analysis has indicated that polymorphisms in *ERα*, specifically rs2228480, are linked to POP [83]. Additionally, polymorphisms rs2234693 and rs17847075 have also been significantly associated with POP in the Chinese population [78].

Overall, the evidence for the association of polymorphisms in *ER* with POP is still insufficient. Although numerous SNP loci associated with POP have been identified across various ethnic groups, the reliability of these findings necessitates further verification. This underscores the multiplicity of factors influencing POP as a polygenic disease. Furthermore, case–control studies that evaluate SNPs in multiple candidate genes are often limited by factors such as small sample sizes and low accuracy of testing methods.

### 4.3. Whole Exome Sequencing (WES)

WES refers to the sequencing of the entire coding region of a gene with high chromosomal coverage and accuracy, which is extensively applied to detect mutations causing diseases. Rao et al. initially conducted WES on peripheral blood DNA samples from eight patients with POP, identifying two missense variants: c.2668 G>A in *WNK1* (p.G890R) and c.6761 C>T in another gene (p.P2254L). These variants were not present in 231 healthy controls. Additionally, functional experiments revealed that fibroblasts in the uterosacral ligament from POP patients with *WNK1* mutations displayed a looser and more irregular arrangement than those from healthy controls [93]. However, due to the small sample size, the conclusions of this study were limited.

### 4.4. Genome-Wide Association Studies (GWAS)

GWAS employs a case–control association design where SNP loci are identified in both the case group and normal controls. The SNPs are analyzed on a genome-wide scale, allowing for the calculation of variant allele frequencies [94]. Allen-Brady et al. performed the first GWAS analysis of 115 patients with a family history of POP by using Illumina 550K. They identified significant associations with POP at six loci, 4q21, 8q24, 9q22, 15q11, 20p13, and 21q22, in the Dutch population, confirming the role of genetic factors in POP development [95]. Subsequently, Giri et al. performed a GWAS on African– and Hispanic–American women, involving 1274 normal controls and 1427 patients with varying stages of POP. They identified several potential susceptibility loci, including *CPE*, *AL132709.5*, *DPP6*, *PGBD5*, and *SHC3* [96].

Recently, several teams have conducted GWAS analysis for POP with larger sample sizes. Olafsdottir et al. analyzed data from the United Kingdom, comprising 15,010 cases and 340,734 controls. They identified eight sequence variants at seven loci associated with POP (*p* < 5 × 10^−8^), further suggesting a role for connective tissue metabolism and estrogen in POP pathogenesis [97]. The advancement of gene chip technology has facilitated genetic screenings on large scales. Utilizing Illumina Human CoreExome microarrays, Cox et al. detected associations between 20 million SNPs and POP from 1329 patients and 16,383 normal controls. Additionally, four previously reported suspect SNPs were replicated, reinforcing the genetic contribution to POP [98]. A meta-analysis synthesizing GWAS results from 28,086 POP patients and 546,291 controls identified 26 genome-wide significant loci (*p* < 5 × 10^−8^), which included 19 novel loci and 7 previously reported ones. This study demonstrated correlations between several candidate genes and POP, including *LOXL1*, *WNT4*, *EFEMP1*, *FAT4*, *IMPDH1*, *TBX5*, and *SALL1*, alongside new candidate genes such as *CHRDL2*, *ACADVL*, and *PLA2G6*. It further revealed that molecular alterations in connective tissues and abnormalities in urogenital development are crucial in the pathogenesis of POP [99].

The above GWAS results suggest that connective tissue abnormalities are associated with POP. Future research involving larger sample sizes and diverse racial groups will elucidate the genetic predispositions to POP further.

## 5. Epigenetic Studies of POP

In addition to genetic predisposition, environmental influences are undoubtedly crucial in the development of POP. Under environmental effects, the coding pattern of DNA undergoes hereditary alterations, known as epigenetic inheritance. Such mechanisms induce inheritable cellular changes without altering the DNA sequence, including DNA methylation, histone modification, microRNA (miRNA) expression, and DNA replication timing, leading to selective gene expression or repression. Recent studies have indicated that epigenetic mechanisms may partially contribute to the pathogenesis of POP [100].

### 5.1. DNA Methylation

DNA methylation, the most prevalent epigenetic mechanism, is influenced by environmental risk factors. It has been reported that the methylation of promoter regions can potentially inhibit LOX gene expression in women with POP. Researchers observed an 8.2-fold decrease in the LOX expression and increased methylation of CpG sites in tissue from women with grade III prolapse compared to controls [101]. Another study identified differences in the genome-wide methylation in the USL between POP patients and controls through a genome-wide DNA methylation analysis [100]. This study reported 3723 differentially methylated CpG sites, with the gene ontology analysis suggesting an association with the ECM. It was demonstrated that specific key members involved in the ECM metabolism were DNA-methylated. However, due to the limitations imposed by small sample sizes in these studies, these findings require further investigation in future studies.

### 5.2. MicroRNA

MicroRNAs (miRNAs) are small non-coding RNAs that bind to mRNA regulatory sites in target genes, modifying their expression via translational repression or mRNA degradation. One study indicated that miR-221 and miR-222 are upregulated in the USL of women with POP, potentially leading to a reduced ERα expression [102]. Similarly, an increased miR-92 expression and decreased ERβ1 level was observed in POP patients, with the ERβ1 expression being inversely correlated with the miR-92 levels [36]. Additionally, studies have demonstrated that overexpressed miR-30d and miR-181a suppress the expression of HOXA11 in the USLs of POP patients. As a crucial transcription factor regulating the collagen metabolism and homeostasis in the USLs, deficient HOXA11 signaling may contribute to the development of POP [103]. Overall, the current research focuses on the correlation between the expression levels of various miRNAs and the development of POP, primarily affecting the expression of ECM-related metabolic genes. The validity of these findings and the specific mechanisms involved require further investigation.

## 6. Conclusions

POP is a benign gynecological disease that seriously affects women’s quality of life. It has a complex etiology and its pathogenesis remains unclear. Numerous studies have indicated the involvement of genetic factors, identifying several candidate genes and SNPs in the metabolism of pelvic floor connective tissue potentially associated with POP. However, the existing genetic research on POP predominantly comprises isolated case–control studies focusing on single populations or small sample sizes, and has yet to pinpoint definitive causative genes and loci. A multicenter GWAS is anticipated to yield more accurate conclusions.

Injury to the levator ani muscles and the dysregulation of the connective tissue component of the pelvic floor may initiate this disorder, or the progression of this disorder itself may lead to molecular changes in the pelvic floor tissue. Patients with POP may possess a genetic predisposition that increases their likelihood of developing pelvic floor dysfunction spontaneously, or as a result of factors such as vaginal delivery, aging, or chronic increased intra-abdominal pressure. Conversely, POP may directly affect gene expression and regulation through tissue stretching. Functional in vitro experiments are essential to further explore the impact of mechanical stress and other factors on the homeostasis of the ECM. Investigating genetic susceptibility and the pathogenesis of POP will aid in providing theoretical recommendations for an individualized risk assessment and diagnosis, as well as the development of targeted therapeutic strategies.

## Figures and Tables

**Figure 1 biomolecules-14-01097-f001:**
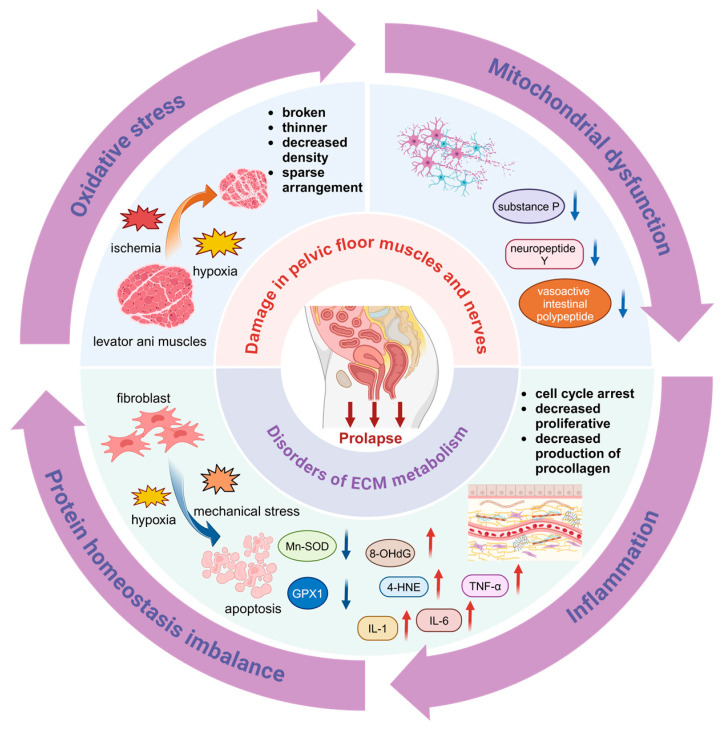
The possible pathogenesis of POP. HIF 1α: hypoxia-inducible factor 1α; GPX1: glutathione peroxidase 1; Mn-SOD: Mn-superoxide dismutase; 8-OHdG: 8-hydroxyguanosine; 4-HNE: 4-hydroxynonenal. IL-1: interleukin 1; IL-6: interleukin 6; TNF-α: tumor necrosis factor α. The straight blue arrows indicate decreased expression levels, and the straight red arrows indicate increased expression levels. Created with BioRender.com.

**Table 1 biomolecules-14-01097-t001:** Analysis of SNPs for POP in ethnically diverse population.

Gene	SNPs	Sample Size (Case/Control)	Population	Correlation to POP	Year	Reference
COL1A1	rs1800012	52/28	Japanese	No	2021	[54]
		348/286	Brazilian	No	2020	[55]
		48/48	Chinese	No	2019	[56]
		36/36	Caucasian, Ashkenazi–Jewish Israeli	No	2009	[57]
	Sp1 G>T	137/96	Italian	No	2012	[58]
		15/15	Korean	No	2009	[59]
		107/209	Brazilian	No	2008	[60]
COL3A1	rs1800255	52/28	Japanese	No	2021	[54]
		348/286	Brazilian	No	2020	[55]
		48/48	Chinese	No	2019	[54]
		112/180	Brazilian	No	2019	[61]
		272/82	Dutch	No	2014	[62]
		202/102	Netherland	Positive	2009	[63]
		84/147	Taiwan	Negative	2008	[64]
	Exon 31, 2092G>A	107/209	Brazilian	No	2011	[65]
		36/36	Korean	Positive	2008	[66]
	rs76425569	48/48	Chinese	Positive	2019	[56]
	rs388222	48/48	Chinese	Positive	2019	[56]
	rs2281968	48/48	Chinese	Positive	2019	[56]
LAMC1	rs10911193(C/T)	239/197	Non-Hispanic white women	No	2012	[67]
		165/246	Caucasian/African–American	No	2010	[68]
		family pedigree	American	Positive	2007	[51]
	rs20563 (A/G)	165/246 239/197	Caucasian/African–American non-Hispanic white women	No	2010	[68]
	rs20558 (T/C)	165/246	Caucasian/African–American	No	2010	[68]
LOXL1	rs2165241	426/410	Brazilian	No	2022	[69]
		52/28	Japanese	No	2021	[54]
LOXL4	rs2862296	52/28	Japanese	Positive	2021	[54]
		285/247	Brazilian	No	2018	[70]
FBLN5	rs2018736	210/292	Russian	Positive	2014	[71]
	rs12589592	210/292	Russian	Positive	2014	[71]
	rs12586948	112/180	Brazilian	No	2020	[72]
MMP1	−1607/−1608,1G/2G	133/132	Poland	No	2013	[73]
		137/96	Italian	Positive	2012	[58]
MMP3	−1612/−1617,5A/6A	133/132	Poland	No	2013	[73]
	−1171 5A/6A	137/96	Italian	No	2012	[58]
	−1171 5A/6A rs3025058	112/180	Brazilian	No	2019	[74]
MMP9	−1562 C/T	137/96	Italian	No	2012	[58]
	rs17576	239/197	non-Hispanic white women	No	2012	[75]
		92/152	Taiwanese	Positive	2010	[76]
MMP10	rs17435959	91/172	Chinese	Positive	2015	[77]
ESR1	rs2234693	88/108	Chinese	Positive	2019	[78]
	rs17847075	88/108	Chinese	Positive	2019	[78]
	rs2228480 G/A	26/26	Israel	Positive	2017	[79]

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
