# Peer review of "Genetics of Female Pelvic Organ Prolapse: Up to Date"

_biomolecules, 2024, doi:10.3390/biom14091097_

Round 1
Reviewer 1 Report
Comments and Suggestions for Authors
The review article submitted by Yuting Li and colleagues provides a comprehensive analysis of pelvic organ prolapse (POP), focusing on its pathogenesis and genetic predisposition. It examines how nerve-muscle damage and extracellular matrix disorders contribute to POP, and highlights recent research linking genetic factors to the condition. The review aims to synthesize current knowledge to enhance risk assessment, diagnosis, and personalized treatment strategies for POP.
Overall, the review paper is well-written. It covers most of the literature relevant to the pathogenesis and genetics of POP. However, I suggest making the following changes to the manuscript before being considered for publication in Biomolecules journals.
1. In addition to oxidative stress, mitochondrial dysfunction, and estrogen deficiency, the following factors also contribute to the complex pathophysiology of POP by disrupting the balance of ECM synthesis and degradation. Including them will enhance the significance of the article.
a. Inflammatory cytokines such as IL-1, IL-6, and TNF-α have been shown to play a role in the pathogenesis of POP by promoting the breakdown of ECM components.
b. An imbalance between proteolytic enzymes MMPs and TIMPs can lead to excessive ECM degradation, contributing to the weakening of pelvic floor tissues observed in POP.
c. Repeated mechanical stress due to factors like childbirth, heavy lifting, or chronic coughing can lead to ECM damage and pelvic floor dysfunction.
d. In addition to estrogen deficiency, other hormonal changes, such as alterations in progesterone levels, can also affect ECM metabolism.
2. The genetic link of the POP has been discussed but it is advisable to include the recent reports on epigenetic mechanisms such as DNA methylation, histone modifications, and small RNAs involved in the pathogenesis of POP.
3. several abbreviations are not explained. Please elaborate all the abbreviations used in the manuscript when appear first in the manuscript.
4. Many gene names are mentioned in the manuscript in normal font, please write all gene names in italic format.
5. Lines 31 to 34, mention where was the study conducted (USA).
6. Lines 79 to 81, change to past tense since findings of a single study are being reported.
7. Line 117, “mitochondrial fusion protein 117 gene 2 (MFN2)” to “mitofusin 2 (MFN2)” or “mitochondrial fusion 2 (MFN2)”
8. 118, change “mediates” to “mediate”
9. Line 213, change “Linkage analysis is a technique used to identify genetic effects” to “Linkage analysis is a technique used to map the location of genes associated with specific traits or diseases by studying inheritance patterns of genetic markers.”
Regards
Author Response
The review article submitted by Yuting Li and colleagues provides a comprehensive analysis of pelvic organ prolapse (POP), focusing on its pathogenesis and genetic predisposition. It examines how nerve-muscle damage and extracellular matrix disorders contribute to POP, and highlights recent research linking genetic factors to the condition. The review aims to synthesize current knowledge to enhance risk assessment, diagnosis, and personalized treatment strategies for POP.
Overall, the review paper is well-written. It covers most of the literature relevant to the pathogenesis and genetics of POP. However, I suggest making the following changes to the manuscript before being considered for publication in Biomolecules journals.
Author Response
Thank you for recognition of our work. We appreciate your time and effort in evaluating our manuscript and raising constructive comments.
Reviewer comment #1
- In addition to oxidative stress, mitochondrial dysfunction, and estrogen deficiency, the following factors also contribute to the complex pathophysiology of POP by disrupting the balance of ECM synthesis and degradation. Including them will enhance the significance of the article.
- Inflammatory cytokines such as IL-1, IL-6, and TNF-α have been shown to play a role in the pathogenesis of POP by promoting the breakdown of ECM components.
- An imbalance between proteolytic enzymes MMPs and TIMPs can lead to excessive ECM degradation, contributing to the weakening of pelvic floor tissues observed in POP.
- Repeated mechanical stress due to factors like childbirth, heavy lifting, or chronic coughing can lead to ECM damage and pelvic floor dysfunction.
- In addition to estrogen deficiency, other hormonal changes, such as alterations in progesterone levels, can also affect ECM metabolism.
Author Response #1
Thank you for your constructive suggestion. We agree with your opinion that including these contents you mentioned will definitely enhance the significance of the article. We have already added a new section to summarize the role of inflammation in the pathogenesis of POP as follows: “Inflammatory cytokines such as IL-1, IL-6, and TNF-α have been implicated in the pathogenesis of POP through the promotion of ECM component breakdown. It has been observed that in patients with POP, the levels of pro-inflammatory cytokines IL-6 and TNF-α positively correlate with MMP1 and MMP2, while inversely correlating with TIMP1 [20,38]. Furthermore, IL-1β plays a role in the regulation of elastin ex-pression [39]. Repeated mechanical stress, resulting from factors such as childbirth, heavy lifting, or chronic coughing, can upregulate genes associated with inflammation, subsequently leading to ECM degradation and pelvic floor dysfunction [37,40]” (Lines 168-176)
As you mentioned, an imbalance between proteolytic enzymes MMPs and TIMPs can lead to excessive ECM degradation, contributing to the weakening of pelvic floor tissues observed in POP. It’s one of the key aspects of ECM metabolism, so we have included these related descriptions as follows: “On the other hand, an imbalance between proteolytic enzymes matrix metalloproteinase (MMPs) and tissue inhibitor of matrix metalloproteinases (TIMPs) can also lead to excessive ECM degradation, contributing to the weakening of pelvic floor tissues observed in POP [20]” (Lines 102-105)
Repeated mechanical stress not only affects pelvic floor muscle and nerve function, but also pelvic floor ECM through chronic inflammation. Related description has been added in our revised manuscript as follows: “Repeated mechanical stress, resulting from factors such as childbirth, heavy lifting, or chronic coughing, can upregulate genes associated with inflammation, subsequently leading to ECM degradation and pelvic floor dysfunction [37,40]” (Lines 174-176)
We added the effects of progesterone on the ECM as follows: “In addition to estrogen deficiency, other hormonal shifts including variation in progesterone levels can influence ECM metabolism. Although findings on progesterone receptor (PR) expression levels are mixed, an increasing number of studies indicate elevated PR in patients with POP [33-35]. Research demonstrates that both estrogen and progesterone contribute to the integrity of pelvic floor support tissues by inhibiting MMP13 and collagenase production, thus reducing ECM degradation [36,37]. It is hypothesized that postmenopausal women experience a loss of this protective effect due to hormone deficiency. Nonetheless, the exact role of progesterone remains unclear” (Lines 159-167)
References:
[20] Cao LL, Yu J, Yang ZL, Qiao X, Ye H, Xi CL, Zhou QC, Hu CC, Zhao CJ, Gong ZL. MMP-1/TIMP-1 expressions in rectal submucosa of females with obstructed defecation syndrome associated with internal rectal prolapse. Histol Histopathol (2019) 34(3):265-274. doi: 10.14670/HH-18-041
[33] Reddy RA, Cortessis V, Dancz C, Klutke J, Stanczyk FZ. Role of sex steroid hormones in pelvic organ prolapse. Menopause (2020) 27(8):941-951. doi: 10.1097/GME.0000000000001546
[34] Fuermetz A. Change of steroid receptor expression in the posterior vaginal wall after local estrogen therapy. Eur J Obstet Gynecol Reprod Biol (2015) 187:45-50. doi: 10.1016/j.ejogrb.2015.02.021
[35] He K, Niu G, Gao J, Liu J-X, Qu H. MicroRNA-92 expression may be associated with reduced estrogen receptor β1 mRNA levels in cervical portion of uterosacral ligaments in women with pelvic organ prolapse. Eur J Obstet Gynecol Reprod Biol (2016) 198:94–99. doi: 10.1016/j.ejogrb.2016.01.007
[36] Zong W, Meyn LA, Moalli PA. The Amount and Activity of Active Matrix Metalloproteinase 13 Is Suppressed by Estra-diol and Progesterone in Human Pelvic Floor Fibroblasts. Biol Reprod (2009) 80(2):367-74. doi: 10.1095/biolreprod.108.072462
[37] Zong W, Jallah ZC, Moalli PA. Repetitive Mechanical Stretch Increases Extracellular Collagenase Activity in Vaginal Fi-broblasts. Female Pelvic Med Reconstr Surg (2010) 16(5):257-262. doi: 10.1097/SPV.0b013e3181ed30d2
[38] Zhou C, Wu Y, Wan S, Lou L, Gu S, Peng J, Zhao S, Hua X. Exosomes isolated from TNF ‐α‐treated bone marrow mes-enchymal stem cells ameliorate pelvic floor dysfunction in rats. J Cell Mol Med (2024) 28:e18451. doi: 10.1111/jcmm.18451
[39] Zhao B, Sun Q, Fan Y, Hu X, Li L, Wang J, Cui S. Transplantation of bone marrow-derived mesenchymal stem cells with silencing of microRNA-138 relieves pelvic organ prolapse through the FBLN5/IL-1β/elastin pathway. Aging (2021) 13(2):3045-3059. doi: 10.18632/aging.202465
[40] Vashaghian M. Gentle cyclic straining of human fibroblasts on electrospun scaffolds enhances their regenerative potential. Acta Biomater (2019) 84:159-168. doi: 10.1016/j.actbio.2018.11.034
Reviewer comment #2
The genetic link of the POP has been discussed but it is advisable to include the recent reports on epigenetic mechanisms such as DNA methylation, histone modifications, and small RNAs involved in the pathogenesis of POP.
Author Response #2
Thank you for your constructive question. We have added the related epigenetic mechanisms of POP in our revised manuscript as follows: “In addition to genetic predisposition, environmental influences are undoubtedly crucial in the development of POP. Under environmental effects, the coding pattern of DNA undergoes hereditary alterations, known as epigenetic inheritance. Such mechanisms induce inheritable cellular changes without altering the DNA sequence, including DNA methylation, histone modification, microRNA (miRNA) expression, and DNA replication timing, leading to selective gene expression or repression. Recent studies have indicated that epigenetic mechanisms may partially contribute to the pathogenesis of POP [99].
4.1 DNA methylation
DNA methylation, the most prevalent epigenetic mechanism, is influenced by environmental risk factors. It has been reported that methylation of promoter regions can potentially inhibit LOX gene expression in women with POP. Researchers observed an 8.2-fold decrease in LOX expression and increased methylation of CpG sites in tissue from women with grade III prolapse compared to controls [100]. Another study identified differences in genome-wide methylation in the USL between POP patients and controls through genome-wide DNA methylation analysis [99]. This study reported 3,723 differentially methylated CpG sites, with gene ontology analysis suggesting an association with the ECM. It was demonstrated that specific key members involved in ECM metabolism were DNA methylated. However, due to the limitations imposed by small sample sizes in these studies, these findings require further investigation in future studies.
4.2 MicroRNA
MicroRNAs (miRNAs) are small non-coding RNAs that bind to mRNA regulatory sites in target genes, modifying their expression via translational repression or mRNA degradation. One study indicated that miR-221 and miR-222 are upregulated in USL of women with POP, potentially leading to reduced ERα expression [101]. Similarly, in-creased miR-92 expression and decreased ERβ1 level was observed in POP patients, with ERβ1 expression being inversely correlated with miR-92 levels [35]. Additionally, studies have demonstrated that overexpressed miR-30d and miR-181a suppress the expression of HOXA11 in the USLs of POP patients. As a crucial transcription factor regulating collagen metabolism and homeostasis in the USLs, deficient HOXA11 sig-naling may contribute to the development of POP [102]. Overall, current research fo-cuses on the correlation between the expression levels of various miRNAs and the de-velopment of POP, primarily affecting the expression of ECM-related metabolic genes. The validity of these findings and the specific mechanisms involved require further investigation.” (Lines: 452-485)
However, we did not retrieve other epigenetics-related studies, including histone modification. Given the complex pathogenesis of POP, research in this area has paid less attention to epigenetic inheritance, which is one of the main focuses for future studies.
References:
[35] He K, Niu G, Gao J, Liu J-X, Qu H. MicroRNA-92 expression may be associated with reduced estrogen receptor β1 mRNA levels in cervical portion of uterosacral ligaments in women with pelvic organ prolapse. Eur J Obstet Gynecol Reprod Biol (2016) 198:94–99. doi: 10.1016/j.ejogrb.2016.01.007
[99] Zhang L, Zheng P, Duan A, Hao Y, Lu C, Lu D. Genome-wide DNA methylation analysis of uterosacral ligaments in women with pelvic organ prolapse. Mol Med Rep (2019) 19:391–399. doi: 10.3892/mmr.2018.9656
[100] Klutke J, Stanczyk FZ, Ji Q, Campeau JD, Klutke CG. Suppression of lysyl oxidase gene expression by methylation in pelvic organ prolapse. Int Urogynecol J (2010) 21:869–872. doi: 10.1007/s00192-010-1108-2
[101] Shi Z, Zhang T, Zhang L, Zhao J, Gong J, Zhao C. Increased microRNA-221/222 and decreased estrogen receptor α in the cervical portion of the uterosacral ligaments from women with pelvic organ prolapse. Int Urogynecol J (2012) 23:929–934. doi: 10.1007/s00192-012-1703-5
[102] Jeon MJ, Kim EJ, Lee M, Kim H, Choi JR, Chae HD, Moon YJ, Kim SK, Bai SW. Micro RNA ‐30d and micro RNA ‐181a regulate HOXA 11 expression in the uterosacral ligaments and are overexpressed in pelvic organ prolapse. J Cell Mol Med (2015) 19:501–509. doi: 10.1111/jcmm.12448
Reviewer comment #3
several abbreviations are not explained. Please elaborate all the abbreviations used in the manuscript when appear first in the manuscript.
Author Response #3
Thank you for your kind reminds. We have elaborated all the abbreviations when they first appear in our revised manuscript.
Reviewer comment #4
Many gene names are mentioned in the manuscript in normal font, please write all gene names in italic format.
Author Response #4
Thank you for your delightful suggestion. We have modified all gene names in italic format in our revised manuscript.
Reviewer comment #5
Lines 31 to 34, mention where was the study conducted (USA).
Author Response #5
Thank you for your kind reminds. We have clearly stated where the study was being conducted in our revised manuscript as follows: “A National Health and Nutrition Examination Survey conducted in America from 2005 to 2010”.(Line 33)
Reviewer comment #6
Lines 79 to 81, change to past tense since findings of a single study are being reported.
Author Response #6
Thank you for your delightful suggestion. We have modified the past tense in our revised manuscript as follows: “women with POP frequently exhibited defects in the levator ani muscles and produce less vaginal closure force during maximal contraction”.(Line 68)
Reviewer comment #7
Line 117, “mitochondrial fusion protein gene 2 (MFN2)” to “mitofusin 2 (MFN2)” or “mitochondrial fusion 2 (MFN2)”
Author Response #7
Thank you for your delightful suggestion. We have modified the “mitochondrial fusion protein gene 2 (MFN2)” to “mitochondrial fusion 2 (MFN2)” in our revised manuscript.
Reviewer comment #8
118, change “mediates” to “mediate”
Author Response #8
Thank you for your delightful suggestion. We have modified the “mediates” to “mediate” in our revised manuscript.
Reviewer comment #9
Line 213, change “Linkage analysis is a technique used to identify genetic effects” to “Linkage analysis is a technique used to map the location of genes associated with specific traits or diseases by studying inheritance patterns of genetic markers.”
Author Response #9
Thank you for your delightful suggestion. We have modified “Linkage analysis is a technique used to identify genetic effects” to “Linkage analysis is a technique used to map the location of genes associated with specific traits or diseases by studying inheritance patterns of genetic markers” in our revised manuscript (Lines: 214-215).

Reviewer 2 Report
Comments and Suggestions for Authors
The manuscript must be substantially modified as it is not at all acceptable as it stands.
First of all, there are many reviews on the pathogenesis of prolapse, cited countless times (one of the most cited, for example, is PMID: 19629013).
To avoid repetitions, the title should be changed, for example, to:
Genetics of female pelvic organ prolapse: up to date.
In this way we focus on a specific review on the genetics of prolapse.
Then the entire first introductory part on the pathogenesis of prolapse (already repeatedly cited in the literature) must be eliminated, reducing it to a minimum, i.e. starting from point 3: Genetic studies of POP.
Finally, we must focus on the genetic specifications of patients suffering from genital prolapse.
Please remember that the review is a narrative review and must be specified both in the abstract and in the text.
The abstract should be revised with a better structure and specification of the points of interest of the manuscript.
Comments on the Quality of English LanguageSome passages in the text need to be reviewed.
Author Response
Reviewer comments:
The manuscript must be substantially modified as it is not at all acceptable as it stands.
First of all, there are many reviews on the pathogenesis of prolapse, cited countless times (one of the most cited, for example, is PMID: 19629013IF: 2.8 Q1 ).
To avoid repetitions, the title should be changed, for example, to:
Genetics of female pelvic organ prolapse: up to date.
In this way we focus on a specific review on the genetics of prolapse.
Then the entire first introductory part on the pathogenesis of prolapse (already repeatedly cited in the literature) must be eliminated, reducing it to a minimum, i.e. starting from point 3: Genetic studies of POP.
Finally, we must focus on the genetic specifications of patients suffering from genital prolapse.
Please remember that the review is a narrative review and must be specified both in the abstract and in the text.
The abstract should be revised with a better structure and specification of the points of interest of the manuscript.
Author Response:
Thank you for your constructive suggestions. We have made point-to-point revision as follows:
(1) As you said, there are some reviews on the pathogenesis of prolapse. We have carefully read the review you mentioned (PMID: 19629013), which goes into great detail about the etiology and risk factors for POP and summarizes the biological changes that occur in the muscles, nerves and ligaments of the pelvic floor. We have added it as a reference in our revised manuscript. As the etiology and risk factors have been more fully summarized and described in previous reviews, we have deleted the corresponding sections in our revised manuscript and focused on the molecular mechanisms underlying the development of POP, which have been less frequently summarized in published reviews.
(2) We understand and agree with your concerns. Given that more space throughout the review is focused on exploring research advances in genetic studies of POP, we have revised the title of the article to “Genetics of female pelvic organ prolapse: up to date”.
(3) After a detailed literature research, we eliminated the introduction of some basic concepts and knowledge of pelvic floor anatomy that might be repeated in the pathogenesis of POP. The revised manuscript summarized the pathophysiological processes and related molecular mechanisms that might be associated with the pathogenesis of POP in a more concise and clearer manner. We initially intended to provide a review of POP-related genetic studies, but when we collated and summarized the information, we found that most of the genetic studies focused on genes related to pelvic floor extracellular matrix metabolism. In order to provide readers with a more in-depth understanding of the content of this part, we summarized the molecular mechanisms related to POP pathogenesis, which is considered to be a pavement to the later part of the content. As you said, the focus of this paper should be placed on the genetic susceptibility. Therefore, integrating the comments of another reviewer, we have made the necessary deletion of this contents and added specific molecular mechanisms.
(4) According to the layout requirements of the journal of “Biomolecules”, we have refined the abstract section to better highlight the genetic studies involved

Round 2
Reviewer 2 Report
Comments and Suggestions for Authors
The authors understood how they needed to edit the manuscript, and did so excellently.
But they forgot to describe the M&Ms.
That is, they did not describe how they wrote the review, what type of searches they performed, what keywords, on which search engines, etc.
A paragraph about M&Ms needs to be added.
Then the manuscript is acceptable for publication.
Author Response
Reviewer:
The authors understood how they needed to edit the manuscript, and did so excellently.
But they forgot to describe the M&Ms.
That is, they did not describe how they wrote the review, what type of searches they performed, what keywords, on which search engines, etc.
A paragraph about M&Ms needs to be added.
Then the manuscript is acceptable for publication.
Author Response
Thank you for recognition of our work! We really appreciate your time and effort in evaluating our manuscript and raising constructive comments. We have added a paragraph about Materials and Methods in our revised manuscript as follows: “A comprehensive search of the published English-language literature was conducted on PubMed. The search was performed using the MeSH terms combined with the following keywords: “pelvic organ prolapse”, “prolapse”, “genetics”, “polymorphism”, “SNP”, “genome wide association study”, “GWAS”, “exome sequencing”, “pathogenesis”, “molecular mechanism”, “muscle”, “nerve”, “extracellular matrix”, “cytokine”, “inflammation”, “senescence”, “aging”, “epigenetic”. After the initial evaluation of titles and abstracts, we included original articles and reviews related to the genetic susceptibility and molecular mechanisms of POP pathogenesis. The references in the retrieved publications were hand searched for relevant studies. Then, two authors independently extracted and made quality judgements on the data, and a third author was consulted in case of disagreement. The quality evaluation system mainly included relevance to genetic studies and molecular mechanisms of pathogenesis of POP, as well as methodological issues.” (Lines: 41-53)